# Assessment of Remotely Sensed Near-Surface Soil Moisture for Distributed Eco-Hydrological Model Implementation

**Carlos Echeverría [1],\*, Guiomar Ruiz-Pérez [2], Cristina Puertes [1] , Luis Samaniego [3] , Brian Barrett [4] and Félix Francés [1]**

[1] Research Group of Hydrological and Environmental Modelling (GIHMA), Research Institute of Water and Environmental Engineering (IIAMA), Universitat Politècnica de València, 46022 Valencia, Spain; cripueca@cam.upv.es (C.P.); ffrances@hma.upv.es (F.F.)

[2] Department of Crop Production Ecology, Swedish University of Agricultural Sciences, 750 07 Uppsala, Sweden; guimar.ruiz.perez@slu.se

[3] Department of Computational Hydrosystems, Helmholtz Centre for Environmental Research—UFZ, 04318 Leipzig, Germany; luis.samaniego@ufz.de

[4] School of Geographical and Earth Sciences, University of Glasgow, Glasgow G12 8QQ, UK; Brian.Barrett@glasgow.ac.uk

\* Correspondence: carec@doctor.upv.es; Tel.: +34-963-876-152

**Abstract:** The aim of this study was to implement an eco-hydrological distributed model using only remotely sensed information (soil moisture and leaf area index) during the calibration phase. Four soil moisture-based metrics were assessed, and the best alternative was chosen, which was a metric based on the similarity between the principal components that explained at least 95% of the soil moisture variation and the Nash-Sutcliffe Efficiency (NSE) index between simulated and observed surface soil moisture. The selected alternative was compared with a streamflow-based calibration approach. The results showed that the streamflow-based calibration approach, even presenting satisfactory results in the calibration period (NSE = 0.91), performed poorly in the validation period (NSE = 0.47) and Leaf Area Index (LAI) and soil moisture were neither sensitive to the spatio-temporal pattern nor to the spatial correlation in both calibration and validation periods. Hence, the selected soil moisture-based approach showed an acceptable performance in terms of discharges, presenting a negligible decrease in the validation period (ΔNSE = 0.1) and greater sensitivity to the spatio-temporal variables' spatial representation.

**Keywords:** eco-hydrological modelling; remotely sensed soil moisture; objective-function; spatial correlation

---

## 1. Introduction

The traditional approach to hydrological model calibration is based only on observed streamflow time series at available gauging stations within the studied basins. However, the assessment of streamflow for a river basin provides an aggregated signal and provided a limited information on the behavior of other relevant state variables of the system [1–7]. In order to deal with this, the data obtained from remote sensing satellites has become a key alternative [8–17], significantly increasing the use of this type of information in recent decades for state variables used in ecohydrology [18]. In fact, remote sensing data not only provides temporal information, but also valuable information on spatial dynamics, which can facilitate model calibration considering spatial patterns and the temporal dynamics.

Currently there is a high availability of satellite data, almost in real time, with sufficient spatio-temporal resolutions (30 m–25 km) for ecohydrology in most cases and with a spatial distribution covering the entire earth. Among the sources of satellite information that can be used in ecohydrology, the following stand out: real evapotranspiration [19–22], land surface temperature [23,24], different vegetation indices [25–27], near-surface soil moisture (hereafter SM), [28–32] and more recently total water storage anomaly [33].

Soil moisture plays a key role in the hydrological cycle, due to its influence on many processes that directly or indirectly affect the water balance, such as: vegetation growth, hydraulic properties of the ground, evapotranspiration, runoff generation and the processes of infiltration and deep percolation [13,30,34–37]. Despite their importance, SM in-situ measurements are still uncommon in time and space [38,39]. If we consider their high spatial and temporal variability, which together with the associated costs of operation and maintenance, it leads to the conclusion that in general, generating the necessary amount of observational data in-situ is economically unviable, except in small experimental basins or plots. Therefore, the SM data obtained by remote sensing is presented as a good alternative to be incorporated into the modeling of the hydrological cycle [34,40–42].

The main objective of this study is to test the possibility of calibrating a hydrological model on a river basin using only remotely sensed soil moisture products. To do this, we also address a secondary objective that involves solving how to process this type of information in order to implement and calibrate hydrological models. To test our hypotheses, a semi-arid Mediterranean catchment area was selected with an ephemeral regime (the Rambla de la Viuda) and a spatially distributed model TETIS model was used.

## 2. Methodology

### 2.1. Study Area

The Rambla de la Viuda basin covers an area of 1513 km$^2$. It is located east of the Iberian Peninsula, mostly in the Valencian Community (Figure 1). The altitude ranges from 38 to 1751 m above sea level. From a geological point of view, there is a dominance of permeable limestone formations, with 73% of the drainage basin occupied by karst landscapes [43,44]. Therefore, the deep percolation capacity and permeabilities of the aquifers are very high, as well as transmission losses in the river network being significant.

The main soil types are (1) leptosol (48% of the total area), a very shallow and thin soil that lies on hard rocks, located on slopes and suitable for agriculture, (2) calcisol (33%), formed under arid conditions, alternating between dry and wet periods, which the formation of calcium carbonate, (3) cambisol (19%), mineral soil that is suitable for agriculture [43,44]. About 60% of the total catchment area is covered by natural vegetation, such as shrubs (75%) and forests (25%). The cultivated area covers 35% of the study area, of which 85% are irrigated orchards and citrus fruits, and the remaining 15% are dry crops. Wetlands and water bodies cover 1% of the catchment area's total, while urban areas cover 4% of the total.

The catchment area has a typical Mediterranean climate, with a high variation of precipitation. Summers are typically warm presenting high values of humidity whilst winters are mild. The average annual rainfall is 535 mm, but with high spatial variability, being wetter in the northern part of the catchment. The mean temperature is 11 °C, with a strong altitude gradient, resulting in an annual mean reference evapotranspiration of 1014 mm. All averages were calculated between 1970 and 2015. The combination of the climatology and geology of this drainage basin gives rise to a river network that has an ephemeral regime in its entirety.

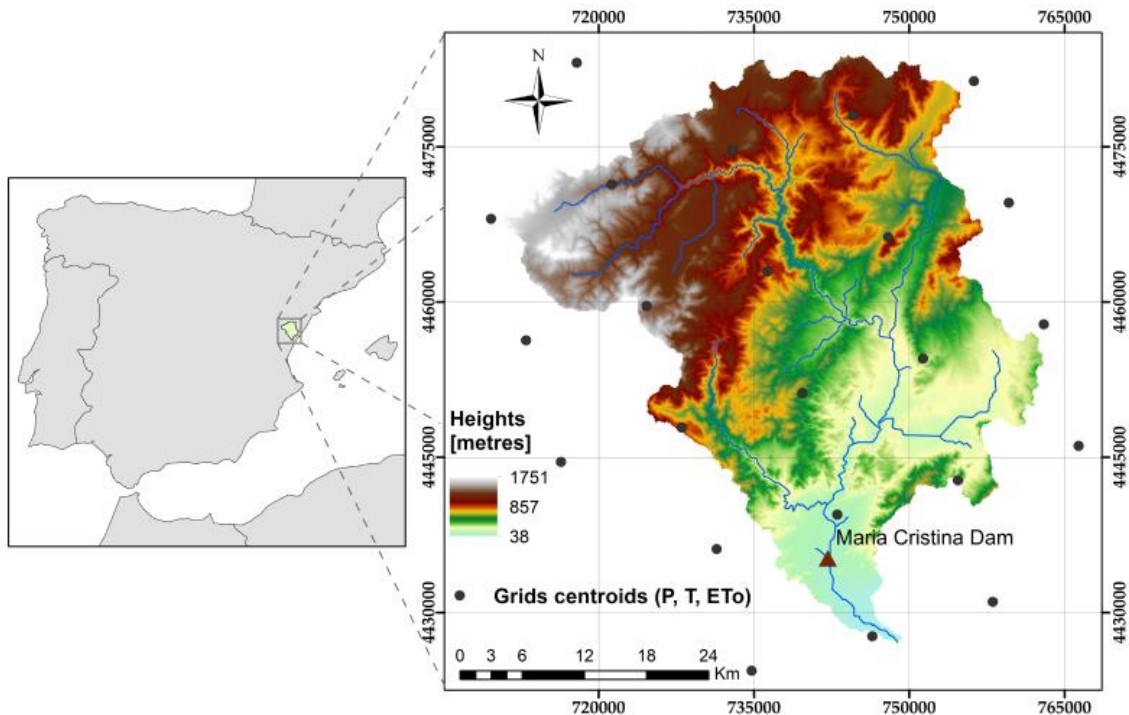

**Figure 1.** Study area: Rambla de la Viuda Catchment with the location of the Spain02 grid points (dots) and the Maria Cristina Dam.

The daily weather data of precipitation and minimum and maximum temperatures come from the reanalysis of all Spain called Spain02, which in version 5 [45] has a spatial resolution of 0.11° (Figure 1) and seasonal coverage from 1950 to 2015. The referenced evapotranspiration was calculated using the Hargreaves-Samani approach [46]. The flows used are those at the entrance to the Maria Cristina reservoir, located near the watershed of the drainage basin (Figure 1), with information from the Center for Studies and Experimentation of Public Works (CEDEX), from 1943 to 2015, available in daily temporal resolution

### 2.2. Satellite Datasets

In respect of the satellite data used as state variables for this investigation, two were used: SM and Leaf Area Index (LAI). On the one hand, the SM was obtained from the Regional Level (L4) surface soil moisture spatial resolution of 1 km, distributed free of charge by the Barcelona Experimental Centre (BEC). This information is the result of the combination of the following products: (1) ESA SMOS L1C brightness temperature, (2) BEC SMOS L3 soil moisture, (3) land surface temperature (LST) provided by European Centre for Medium Weather Forecast (ECMWF), and (4) Normalized Difference Vegetation Index (NDVI) from Moderate Resolution Imaging Spectroradiometer (MODIS) sensor [47,48]. In addition to the SM, the product LAI from the MODIS sensor has been used, which is an eight-day composite dataset with 500 m pixel size [49], also distributed free of charge by NASA. Both products have temporal availability for the studied period of this research, (2010–2015).

### 2.3. Eco-Hydrological Model: TETIS

TETIS is a conceptual model, but uses physically-based, spatially-distributed parameters [50]. This last feature allows users to consider the spatial heterogeneity of inputs, parameters, and state variables.

The TETIS model is based on a series of tanks in each cell, connected both vertically and horizontally, which represent the different hydrological processes and water storages in the vegetation-soil-aquifer column. As we are dealing with an ephemeral river, the underground flow was disabled (Figure 2).

TETIS presents a surface soil layer that specifically models this state variable for humidity below field capacity, thus allowing the use of satellite information on SM [9]. The rest of the soil up to the effective depth of roots and also for SM below field capacity is represented by the static storage. In both cases, the only water flow outlet is by evapotranspiration. The gravitational storage represents the soil to effective depth of roots with moisture from field capacity to saturation (as interflow is possible from this storage). For the propagation along the river network of flows produced on slopes and aquifers, TETIS uses the simplification of the Kinematic Wave and it is possible to introduce transmission losses.

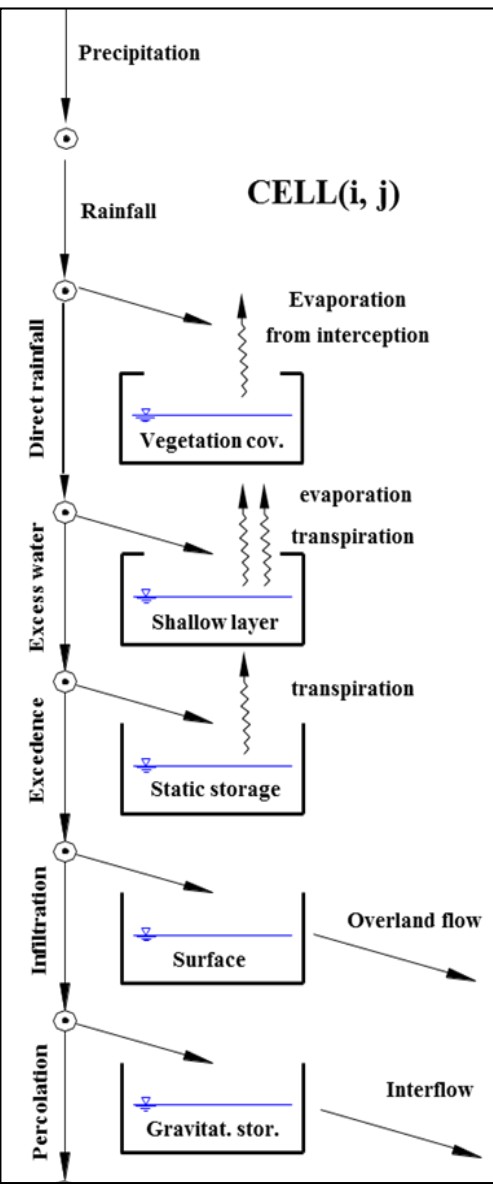

**Figure 2.** TETIS conceptual scheme in a cell, adapted for an ephemeral river. The water content of the shallow layer tank represents the surface SM.

Internally, TETIS presents a separate structure from its effective parameters by using a scalar correction factor for each parameter type. This correction factor is common for all cells and multiplies the map of values of the previously estimated corresponding parameter in each cell [51,52]. This facilitates the calibration process, as the number of variables to be calibrated is limited to the number of corrective factors chosen to be optimized. For this case study, these corrective factors are those that affect: static

storage and maximum interception, evapo-transpiration, infiltration capacity, overland flow velocity, interflow horizontal saturated permeability, percolation capacity and flow rate in the river network.

It is important to note that to perform basin modeling, TETIS can include a dynamic vegetation sub-model. This sub-model [27,53] has 11 parameters (maximum interception storage, effective depth of the first layer, effective depth of the second layer, light use efficiency, coverage factor, distribution of roots, maximum LAI supported by the system, light extinction coefficient, specific leaf area, optimal temperature, respiration rate) for each type of vegetation cover. In the case of this study, it was decided to calibrate the three most influential parameters in the hydrological cycle: maximum interception storage, coverage factor and distribution of roots. Since there are five vegetated soil coverings in the case study (water bodies and urbanized areas are not considered), the end result is a total of 15 vegetation parameters to be calibrated.

The simulation scales with TETIS have daily steps and pixel size of 100 m, generating results in spatial format with minimum resolution of 100 m and minimum temporal resolution of one day. Because SM and LAI have resolutions of 1000 m and 500 m, respectively, a spatial aggregate of the TETIS output information was performed to make a comparison. In the case of the LAI, which also has an 8-day resolution, a temporal aggregation was also carried out.

## 2.4. SM-Based Efficiency Indices

Taking into account the complexity of the dynamics of SM, both spatially and temporally [35,36], as well as the lack of information on the characterization of errors and the uncertainty associated with satellite-obtained SM [9], this research focused on efficiency indices that only took spatial patterns and temporal variability into account rather than modeling the exact values observed by remote sensing at each pixel. For this purpose, the methodology of the Empirical Orthogonal Functions (EOF) was chosen to analyze the data set of the SM. EOFs have been widely used in climate sciences [19,36,54–57] and with recent applications in hydrology [27,58]. Other studies have suggested alternative metrics to evaluate the spatial accuracy of such variables, such as the use of Map Curves [59] or the concordance coefficient [60]. The EOF methodology divides the spatio-temporal dataset into a linear combination of two components: eigenvalues and eigenvectors. The former are invariants in time and the latter in space. These elements are often referred to as "main components" and "loadings", respectively. The main components contain the variance part of the explained total and the loadings contain the variability of orthogonal spatial patterns [36,61–63].

Matrix **M** is a spatio-temporal dataset with dimensions (s, t), where each of the rows s represents a map for a given time, and each column t is the temporal series for each pixel of the spatial division.

The first step is to calculate the anomaly matrix **N** (s, t) (Equation (1)), by subtraction of the spatial mean, denominated **B** (1, t), from **M**, in each temporal series corresponding to the matrix **M**.

$$\mathbf{N} = \mathbf{M} - \mathbf{B}, \tag{1}$$

The covariance matrix **C** (Equation (2)) is then calculated and the equation of eigenvalues is solved. In Equation (3), $\mathbf{\Lambda}$ is a diagonal matrix containing the eigenvalues $\lambda_s$ of the correlation matrix **C**, with the columns of **E** being their corresponding eigenvectors.

$$\mathbf{C} = \mathbf{N}^{\mathbf{T}} \times \mathbf{N}, \tag{2}$$

$$\mathbf{C} \times \mathbf{E} = \mathbf{E} \times \mathbf{\Lambda}, \tag{3}$$

Each of **E**'s eigenvectors can be thought of as a map describing spatial patterns (EOFs). In general, the n eigenvalues and their corresponding eigenvectors are selected, which explain most of the variance, and the dataset is reconstructed as shown in Equation (4), where $a_i$ represents the temporal evolution

of EOF$_i$, which will be referred to hereafter as loadings. Generally, the number of eigenvalues selected to rebuild the dataset (n) is less than the number of time series(s)

$$\vec{a}_i = \mathbf{N} \times \mathbf{EOF_i} \text{ where } i = 1, \ldots, n; n \ll s, \tag{4}$$

Once the concept of EOF is explained, the SM-based efficiency indices (STEs) will be defined below to be used later as objective functions (OFs) in the process of calibration of the eco-hydrological model. In total, four indices were analyzed, taking as a starting point the one proposed by [64] and employed by [27], considering the number of main components explaining at least 95% of the variance. The formula is described in Equation (5).

$$STE_1 = \sum_{i=1}^{pc} w_i \times \sum_{j=1}^{t} \left( \left| load\_obs_{i,j} - load\_sim_{i,j} \right| \right), \tag{5}$$

where $w_i$ is the explained variance of the EOF methodology for the main component i; pc is the number of main components that explain at least 95% of the variance; $load\_obs_{i,j}$ is the observed SM loading series obtained by the EOF methodology; $load\_sim_{i,j}$ is the simulated SM loading series obtained by the EOF methodology.

In order to give more weight to the pattern rather than the exact value, another complementary EOF was used which is based on the calculation of the NSE index of observed and simulated loadings (Equation (6)).

$$STE_2 = \sum_{i=1}^{pc} \left[ w_i \times NSE(load_i) \right], \tag{6}$$

Another alternative, following the use of the NSE index, was to calculate the index applied to the observed and simulated SM at each pixel throughout the period analyzed and to use the value obtained in each pixel to distinguish between pixels in which the model showed good performance and those in which it did not. The criterion used to carry out this distinction was that the NSE was greater than or equal to 0.5. Once the pixels were classified according to this criterion, the ratio between the number of pixels that meet the condition was divided by the total number of pixels, getting the index named STE$_3$.

$$STE_3 = \frac{1}{p} \times \sum_{i=1}^{p} I_{[u,1]} \left[ NSE(SM_i) \right] \forall u \geq 0.5 \tag{7}$$

Finally, evaluation of the performance of the SM as a state variable was proposed using a formula that calculates the sum of the NSE index applied to the SM in each pixel for the period analyzed for each major component, divided by the number of components explaining at least 95% of the variance. This formula gives equal weight to all principal components.

$$STE_4 = \frac{1}{pc} \times \sum_{i=1}^{pc} NSE(load_i) \tag{8}$$

### 2.5. Optimisation Algorithm and Goodness-of-Fit Indexes

The automatic optimization algorithm Shuffled Complex Evolution was selected (SCE-UA) [65] to perform the automatic calibration of the distributed eco-hydrological model. A total of 22 parameters were calibrated, of which seven correspond to the hydrological sub-model and the remaining 15 correspond to the dynamic vegetation sub-model. The model has a warm-up period comprising the entire year 2010, it was calibrated for the period 2011–2013 and it was validated for the period 2014–2015.

In total, five calibrations were performed. First, four SM-based calibration alternatives were proposed, which were compared to each other, in order to choose the best STE index. Second, the

selected STE index was compared to the streamflow-based configuration, in which the NSE index applied to observed and simulated flows was optimized. In order to make an objective comparison of the results obtained with each of the calibration strategies, the following were calculated for each strategy, both in the calibration and validation period: (a) the NSE index between observed and simulated flows, (b) the NSE index applied to an area mean of the value of all pixels in the drainage basin every 8 days for the entire period under consideration, for both the LAI and the SM, and (c) the Pearson correlation coefficient, (hereafter R), in each pixel for the period analysed, also for the LAI (at 500 m resolution) and SM (at 1000 m resolution).

## 3. Results and Discussion

### 3.1. Selection of SM-Based OF

As a first step, the four SM-based efficiency index proposals have been calibrated and validated, calculating the NSE index of: (a) to discharges into Maria Cristina, (b) the eight-day area average of the SM, (c) to the eight-day area average of the LAI. In addition, the value of the eigen statistics has been calculated, although, they cannot be comparable to each other because they do not have the same range of values or units. However, they can give us an idea of the cross performance and, more importantly, the degradation of performance from calibration to validation.

All SM-based alternatives provided satisfactory results (Figure 3), but $STE_2$ was selected, as it was the most robust in the transition from calibration to the validation period. In addition, it showed the best value and the highest robustness in the NSE index for flow-rates, in addition to performing well in LAI validation for the calibration period, which, despite being a negative value, is the best among all proposals. The behavior of the LAI is quite striking, especially within the calibration period, because it has very negative values, except in option 2, but has a noticeable uptick in the validation period. The negative NSE in the LAI can be explained by a low temporal variability (see for example Figure 3) and a possible initial condition problem in the calibration period.

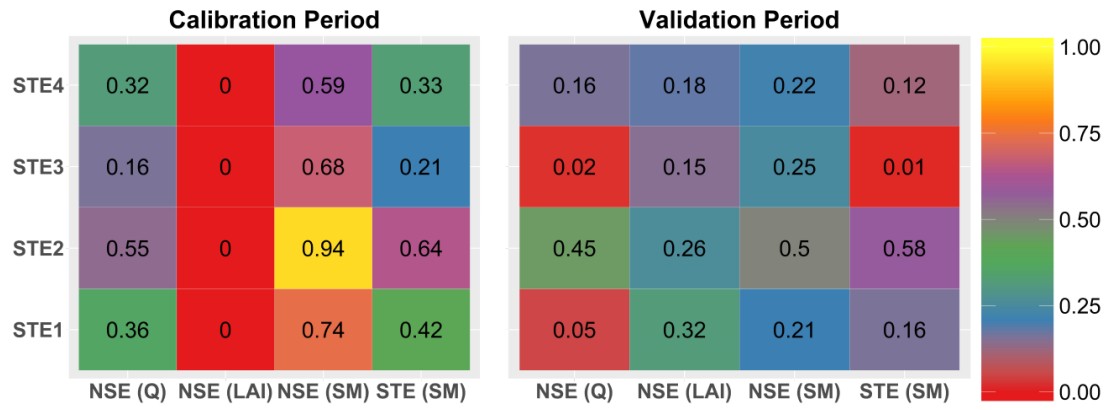

**Figure 3.** Comparison of the proposed soil moisture (SM)-based objective functions (OFs) in calibration and validation periods for different efficiency indexes.

Once the best SM-based configuration was chosen, $STE_2$, a comparison was made between the streamflow-based configuration (NSE index) and the SM-based alternative selected in order to analyze the main advantages and disadvantages of using only the SM as the only variable in the calibration step of the distributed ecohydrological model.

### 3.2. Calibration Period

Given that the OF used in the streamflow-based configuration was precisely the NSE between observed and simulated flows, this configuration presents an NSE of 0.91, much larger than that corresponding to the SM-based configuration (0.55). Although still presenting better results, this

superiority of the streamflow-based configuration over the SM-based is better when comparing the flow duration curve (Figure 4b,d).

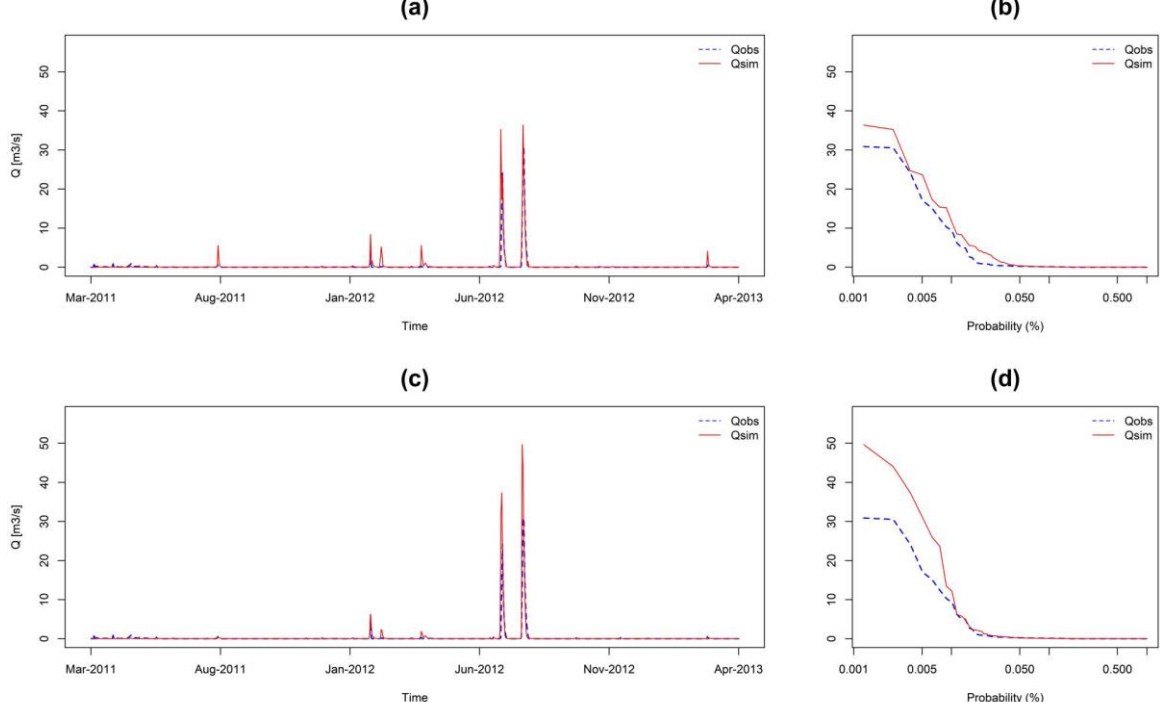

**Figure 4.** Observed and simulated discharges during the calibration period (2011–2013) for streamflow-based configuration (**a**,**b**) and SM-based (**c**,**d**): hydrographs on the left (**a**,**c**) and flow duration curves on the right (**b**,**d**).

The mean area SM of all pixels was examined. For the temporal dynamics of the SM for the calibration period in the streamflow-based configuration (Figure 5a), the simulated values generally followed the pattern of the observed values. For the SM-based configuration (Figure 5b), the simulated time series were much more sensitive to the observed values.

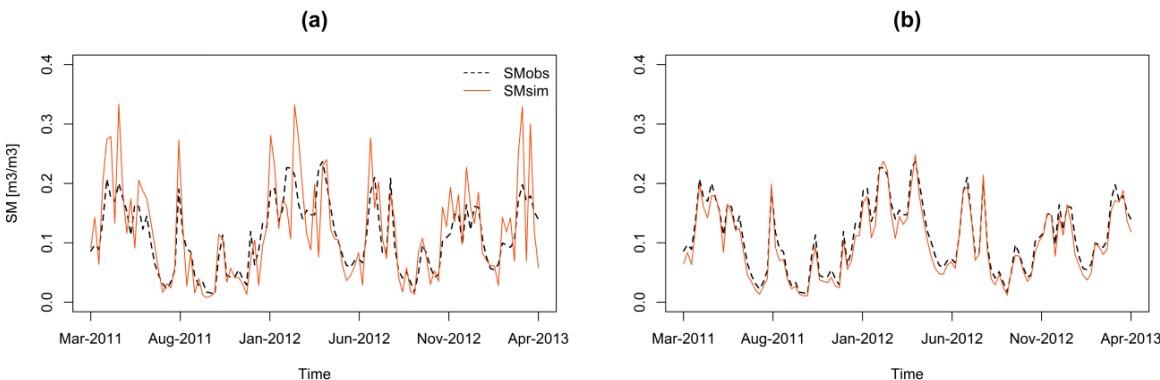

**Figure 5.** Observed and simulated mean area SM during the calibration period (2011–2013) for: (**a**) streamflow-based configuration; (**b**) SM-based configuration.

With respect to the average LAI for the calibration period, in the case of the streamflow-based approach (Figure 6a), the simulated LAI always reproduced the pattern of the observed values, but with a more marked oscillation. For SM-based approach (Figure 6b), the differences between the simulated and observed series were smaller than in the streamflow-based approach and the simulated pattern was more similar to the observed pattern.

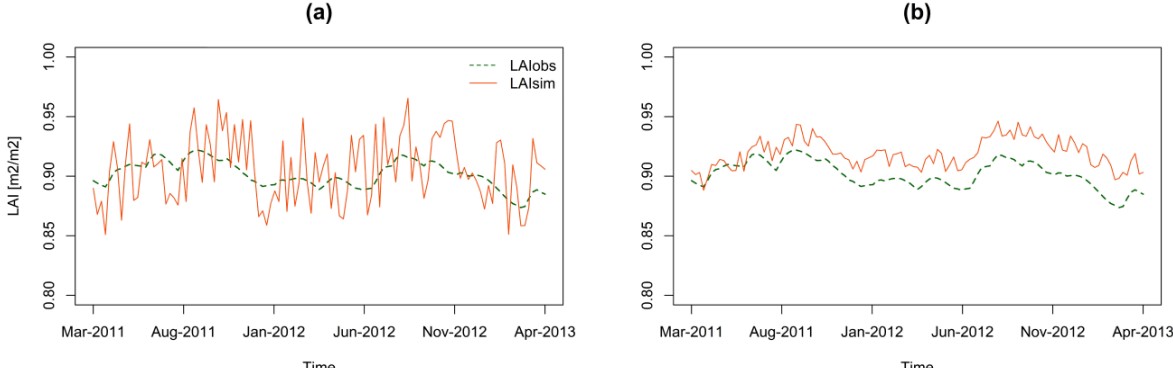

**Figure 6.** Observed and simulated mean area LAI during the calibration period (2011–2013) for: (**a**) streamflow-based configuration; (**b**) the SM-based configuration.

In relation to the spatial distribution of R for SM, with the streamflow-based configuration (Figure 7a) the coefficient mainly had values between 0 and 0.20 (approximately 45% of the total area) whilst 35% of the values were between −0.20 and 0, and in a smaller proportion it was possible to observe R values of less than −0.20 (20%). That is, no pixel with correlation greater than 0.20 was present, and 55% had negative values. For the SM-based configuration (Figure 7b), R was almost always greater than 0.40, except for 3 pixels with a value between 0.30 and 0.35, with the area almost completely dominated (70%) by cells with the highest values of the scale (above 0.80), while 25% of the cells had values between 0.60 and 0.80 and only around 5% between 0.40 and 0.60.

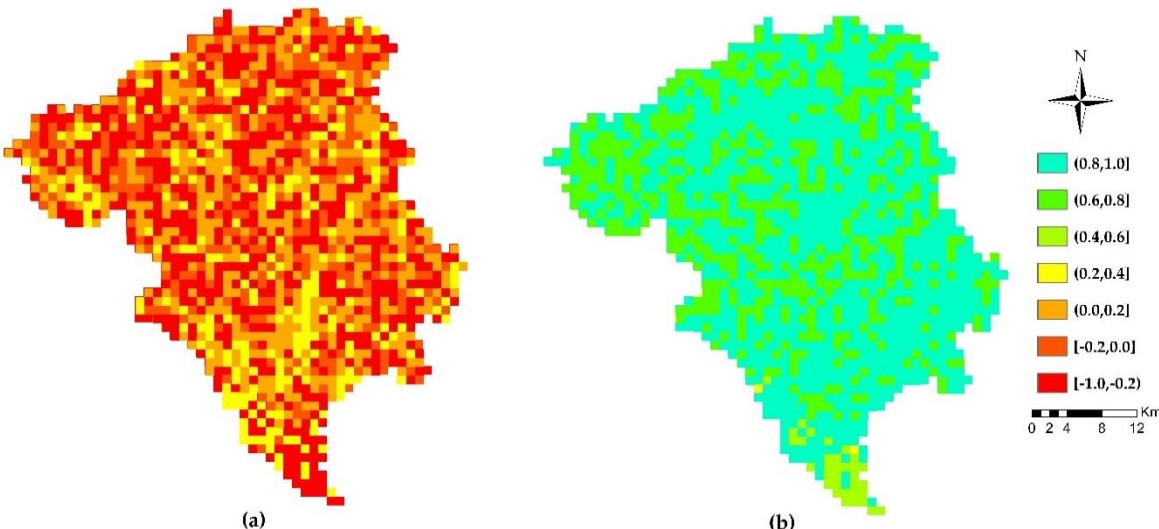

**Figure 7.** Spatial distribution of R for SM during calibration period (2011–2013) for: (**a**) streamflow-based configuration; (**b**) SM-based configuration.

For the LAI case, the R values between the simulated and observed variables varied between −0.20 and 0.20 in the streamflow-based configuration (Figure 8a), covering approximately the same area (49% and 51%). For the SM-based configuration (Figure 8b), R was higher in the North region, with values between 0.20 and 0.80, with only a few cells between 0.20 and 0.40 (15%) and approximately the same proportion of cells with R values between 0.40 and 0.60 (42%) and R values between 0.60 and 0.80 (43%).

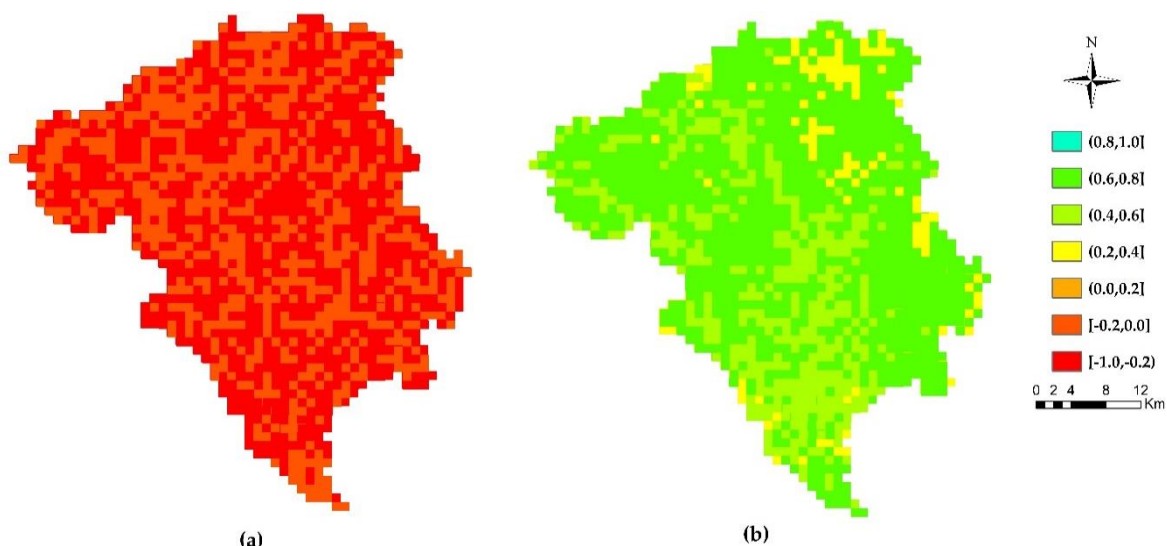

**Figure 8.** Spatial distribution of R for Leaf Area Index (LAI) for the calibration period (2011–2013) for: (**a**) the streamflow-based configuration; (**b**) the SM-based configuration.

*3.3. Validation Period*

In terms of discharge, for the validation period (Figure 9), both configurations performed well following the observed discharge trend, although the SM-based configuration slightly overestimated the peaks ($\Delta$Qpeak $\approx$ 10 m$^3$/s). For performance indices, the NSE index between observed and simulated flows for both configurations was 0.47 and 0.45, respectively. The STE$_2$ values were 0.03 for the streamflow-based configuration and 0.58 for the SM-based approach.

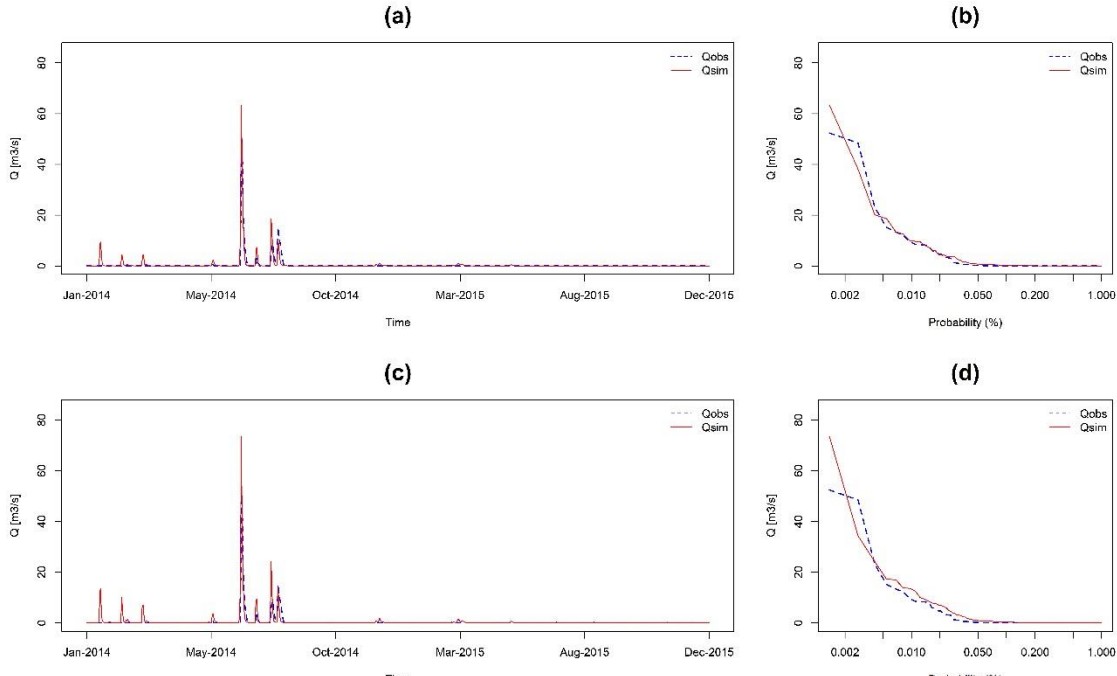

**Figure 9.** Observed and simulated discharges during the validation period (2014–2015) for streamflow-based configuration (**a**,**b**) and SM-based (**c**,**d**): hydrographs on the left (**a**,**c**) and flow duration curves on the right (**b**,**d**).

With regard to the representation of the area mean of the SM, for the streamflow-based configuration, despite reproducing the pattern, the simulated values were generally overestimated

in relation to those observed (Figure 10a). For the SM-based configuration it is possible to observe that, in addition to following the pattern, the values are closer to each other, proving that the latter configuration is more sensitive to this variable (Figure 10b).

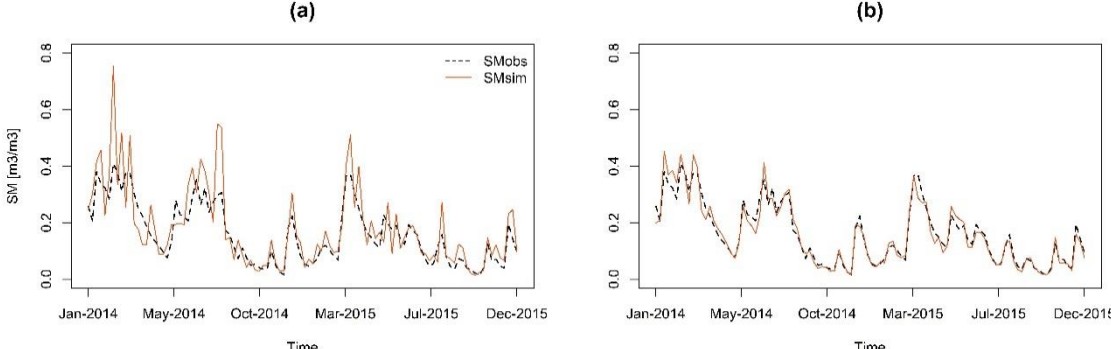

**Figure 10.** Observed and simulated mean areal SM for the validation period (2014–2015): (**a**) for streamflow-based configuration; (**b**) for SM-based configuration.

In the case of the area mean of the LAI for the entire basin, the streamflow-based configuration (Figure 11a) reproduced the observed pattern, but not the exact values, with the simulated values on average being 10% lower than the observed values. In contrast, the SM-based approach (Figure 11b) was able to reproduce satisfactorily, despite small oscillations, the temporal pattern that was followed by the series of observed values of LAI.

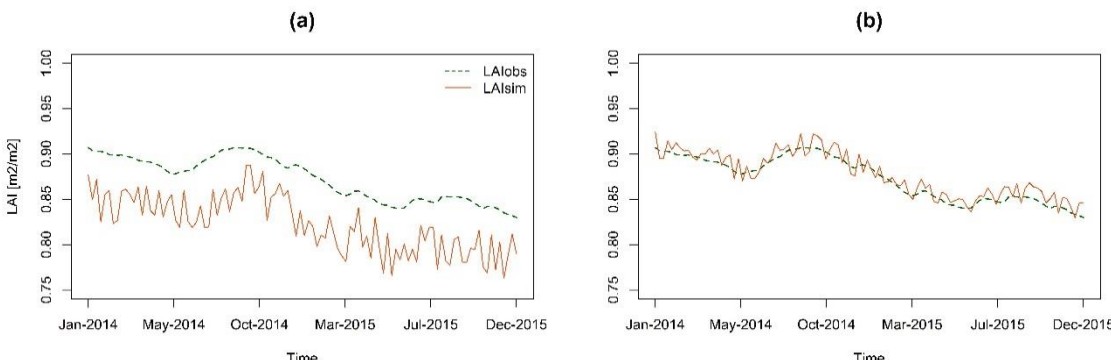

**Figure 11.** Observed and simulated mean areal LAI for the validation period (2014–2015): (**a**) for streamflow-based configuration; (**b**) for SM-based configuration.

The distribution of R was spatially analysed. In the streamflow-based configuration (Figure 12a), the SM (Figure 12a) during the validation period was mostly dominated (approximately 60%) by R values between 0.2 and 0.4, followed by values between 0.0 and 0.2 in a smaller proportion (10%). There were cells with values between −0.2 and 0.0. For the SM-based configuration (Figure 12b), the study area was clearly dominated by R values between 0.4 and 0.8 (80%), with a small presence of R values between 0.2 and 0.4 (20%).

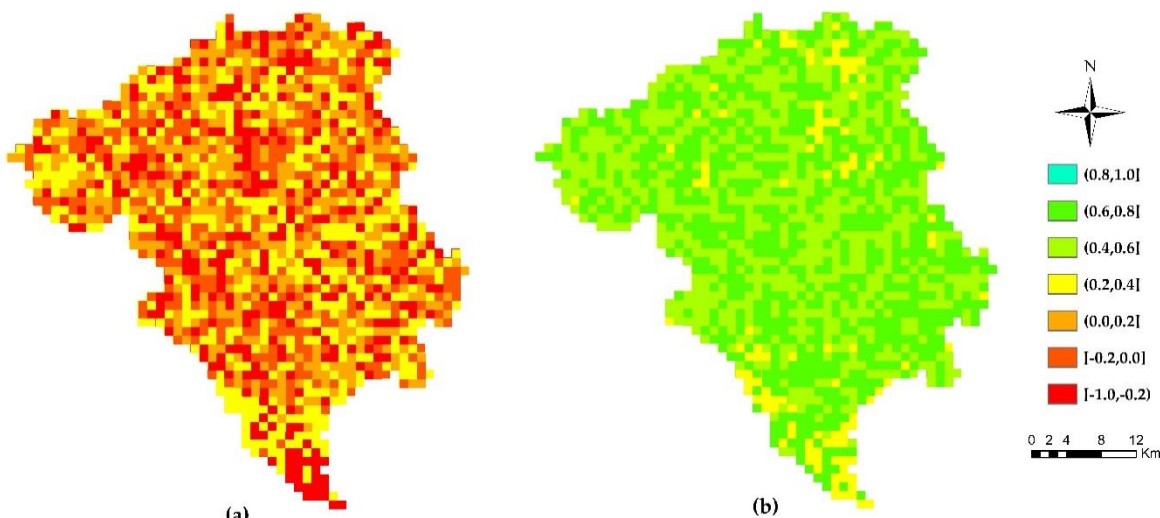

**Figure 12.** Spatial distribution of R for SM for the validation period (2014–2015), for the streamflow-based (**a**) and SM-based (**b**) configurations.

For the LAI in the validation period (Figure 13), with respect to the streamflow-based configuration, the area was dominated by R values less than −0.20 (65%), with complementary presence of R values between −0.20 and 0.40 (35%). For the SM-based configuration, the studied area was dominated by values between 0.40 and 0.80 (60%), with a small presence of values between 0. and 0.40, and only a few cells had negative values.

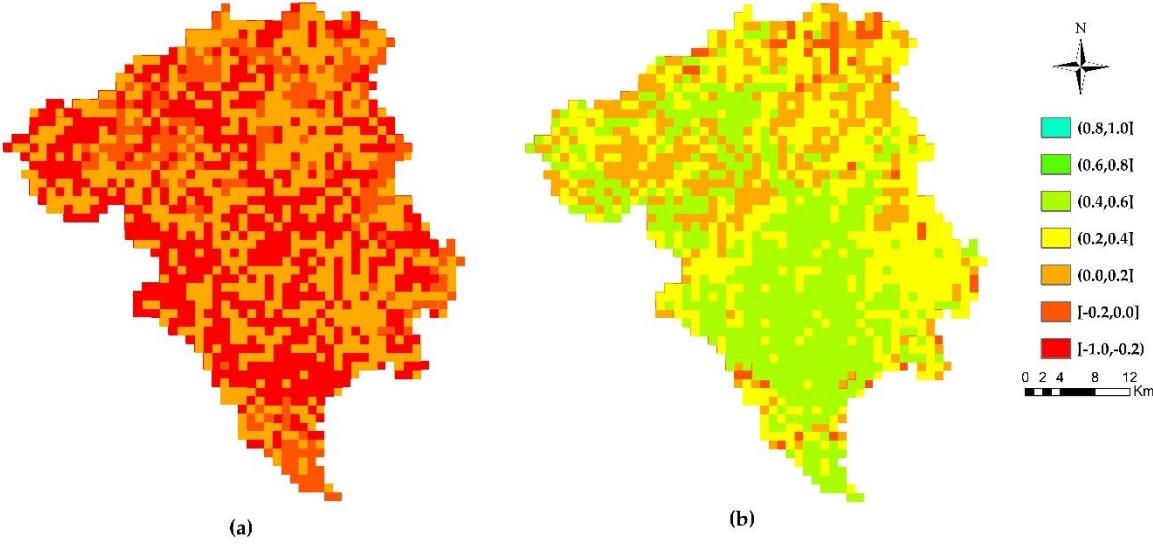

**Figure 13.** Spatial distribution of R for LAI during the validation period (2014–2015), for the streamflow-based (**a**) and SM-based (**b**) configurations.

### 3.4. Value of Satellite Information

The evaluation of a hydrological model using only remotely sensed information is an appropriate alternative for ungauged or data-scarce basins, as opposed to classical methods such as regionalization of parameters, widely used in hydrology for these situations [66–69]. Various authors found clear limitations in the transfer of parameters [70,71], mainly because the estimation of parameters, done in this case, includes information from outside the basin itself [72].

Among other advantages, [30] stated that, through the joint use of spatio-temporal information and discharges, they achieved satisfactory results in the process of optimizing aggregate and

semi-distributed models, drawing possible equifinality situations. This can be clearly observed in this work by a good representation of the area average of spatio-temporal variables (LAI and SM). However, it has not been possible to adequately represent R for the streamflow-based configuration. The same authors have explored optimization with spatio-temporal information of semi-distributed models of some ungauged sub-basins, also with satisfactory results. In this research, calibration was carried out by using a distributed hydrological model, considering the entire study area as ungauged.

The optimization procedure using only temporal information (i.e., discharges at the gauged point) presented good model accuracy at the calibration stage, obtaining an NSE index of 0.91 (Table 1). However, it was not able to represent the SM-based index by getting a low value (i.e., $STE_2$ (SM) = 0.01). During the model validation period, it showed a significant reduction in the time index (NSE(Q) = 0.47) and was unable to obtain a good representation of the SM-based index by presenting a low value ($STE_2$ (SM) = 0.03), even slightly higher than in the calibration period.

**Table 1.** Comparison of the performance of the temporal and SM-based configuration in calibration and validation periods.

| Configuration | Streamflow-Based | | SM-Based | |
|---|---|---|---|---|
| Index | NSE (Q) | STE (SM) | NSE (Q) | STE (SM) |
| Calibration | 0.91 | 0.01 | 0.54 | 0.63 |
| Validation | 0.47 | 0.03 | 0.44 | 0.58 |
| ΔIndex(cal-val) | 0.44 | 0.02 | 0.10 | 0.05 |

In the case of the SM-based configuration (Table 1), at the calibration stage it presented a lower value than expected from the time index (NSE (Q) = 0.54) even though it was acceptable. However, it had an insignificant decrease in the transition to the validation period (NSE (Q) = 0.44). In terms of the $STE_2$ itself, it was also possible to see good stability of the index in the transition from the calibration period to the validation period, also presenting a slight reduction ($STE_2$ (SM) = 0.05).

In addition, it could be seen that considering SM as the state variable in calibration step, did not imply additional dispersion in the process of optimizing the NSE index (Figure 14). The fact that dispersion was not increased by adding remotely sensed information [27,73,74] is an indicator that one does not increase the optimal search space, making the algorithm more efficient in the search for the best solution.

Following the assessment of the value of satellite information for SM calculations, the mean correlation of the whole basin (R) and for the spatial mean SM, the NSE index and the balance error (BE) were obtained. The metrics analyzed showed greater robustness in evaluating the performance of average behavior for SM in the SM-based configuration (Table 2). Thus, for example, these indices have better values and are almost always more stable in the transition of changes in periods, especially in the case of average R, that has almost exactly the same value in both periods.

**Table 2.** SM Comparison of the performance in terms of simulated SM of the streamflow-based and SM-based configuration in calibration and validation periods.

| Configuration | Streamflow-Based | | | SM-Based | | |
|---|---|---|---|---|---|---|
| Index | NSE (SM) | R (SM) | BE (SM) | NSE (SM) | R (SM) | BE (SM) |
| Calibration | 0.63 | −0.12 | −4.20 | 0.94 | 0.71 | −9.50 |
| Validation | 0.39 | 0.11 | −31.00 | 0.50 | 0.70 | −21.40 |
| ΔIndex(cal-val) | 0.24 | 0.23 | 26.80 | 0.44 | 0.01 | 11.90 |

With respect to the LAI, the same statistics as for the SM were calculated, further demonstrating the more pronounced success of the configuration's consideration of this variable (Table 3). In relation to robustness, again the SM-based configuration showed more robustness for all indexes, but more

notable is the large gap of improvement from one configuration to the other. It is also interesting to mention that the mean correlation coefficient for SM (R), as in the previous case, also virtually remains constant in both periods, further emphasizing the robustness mentioned above.

**Table 3.** Comparison of the performance in terms of simulated LAI of the streamflow-based and SM-based configuration in calibration and validation periods.

| Configuration | Streamflow-Based | | | SM-Based | | |
|---|---|---|---|---|---|---|
| Index | NSE (LAI) | R (LAI) | BE (LAI) | NSE (LAI) | R (LAI) | BE (LAI) |
| Calibration | −99.01 | −0.35 | 3.90 | −0.15 | 0.45 | −6.30 |
| Validation | −0.86 | −0.20 | −25.00 | 0.32 | 0.48 | −24.90 |
| ΔIndex$_{(cal-val)}$ | 98.95 | 0.15 | 21.10 | 0.47 | 0.03 | 18.60 |

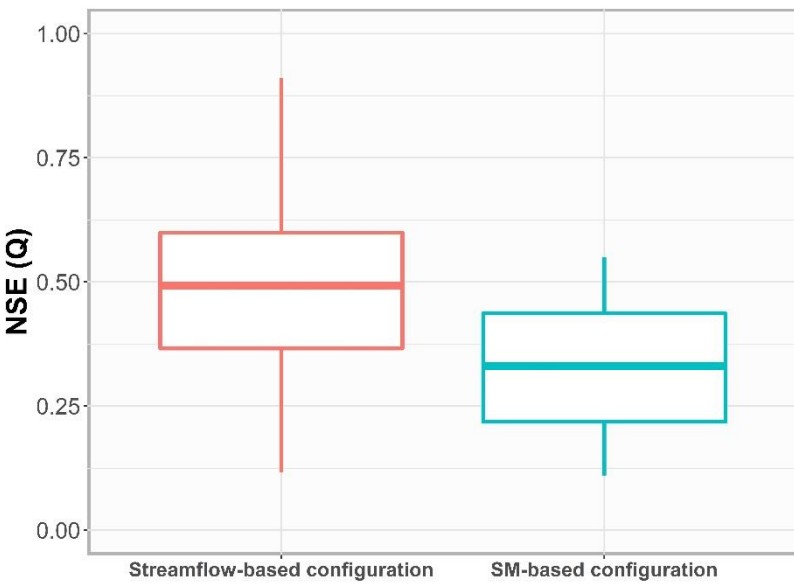

**Figure 14.** Variability of the Nash-Sutcliffe Efficiency (NSE) index between observed and simulated discharges in the search of the optimal result in calibration for streamflow-based and SM-based configurations.

## 4. Conclusions

Four SM-based optimization alternatives based on the EOFs methodology were evaluated. All four had satisfactory results. However, of all the options, the alternative selected was the one that, in general terms, when including the value of the SM-based statistic itself, performed better and showed the greatest robustness in the transition from the calibration period to the period of validation.

The selected SM-based alternative was then compared to the streamflow-based configuration that considers discharges as a state variable, using the NSE index as OF. The two alternatives were evaluated through the analysis of: (a) a hydrograph and flow duration curve for discharges, (b) the temporal evolution of the area mean of SM and LAI for the entire basin, (c) the spatial distribution of the R in the watershed of both SM and LAI.

On one hand, the streamflow-based configuration was not able to spatially represent the pattern of SM and LAI (almost zero obtained for the STE index in both calibration and validation periods). On the other hand, the SM-based configuration was able to achieve good values for the NSE index of the observed and simulated flows: NSE = 0.54 in the calibration period and NSE = 0.44 in the validation period, despite the fact that streamflow data was totally ignored for optimization purposes.

This further demonstrates it exhibits greater robustness in achieving acceptable representation of the hydrological state variable (Q) and SM.

When plotting the area means for the SM and LAI, although the LAI was somewhat difficult to represent in the calibration period, both configurations managed to show satisfactory performance. However, when plotting the spatial distribution of the R for both SM and LAI, the streamflow-based configuration was not able to perform well, whereas the SM-based configuration achieved good representation. This good representation behavior of the area means, but poor representation of spatial correlation in each pixel for the streamflow-based configuration, symbolizes an equifinality situation, which contours with the SM-based configuration.

In addition, the improvement in the robustness of the model was made clear when analysing the dispersion of the NSE index between the observed and simulated flows, since when considering only satellite information for the calibration of the model, no dispersion is added in the optimization of the search space.

Finally, it can be said that the information obtained by remote sensing is a good option for the calibration of eco-hydrological models in basins that are ungauged or with a shortage of flow data. This is because, on the one hand, satisfactory results can be obtained from the point of view of the simulation of flows without using them in the calibration phase and, on the other hand, the robustness of the implemented model can be increased due to the incorporation of the behavior of spatial patterns and temporal dynamics of a state variable observed by remote sensing.

**Author Contributions:** Conceptualization, C.E., F.F. and G.R.-P.; methodology, C.E. and G.R.-P.; software, C.E. and F.F.; validation, C.E., C.P. and B.B.; formal analysis, C.E.; investigation, C.E. and G.R.-P.; resources, C.E., F.F., L.S. and B.B.; data curation, C.E.; writing—original draft preparation, C.E.; writing—review and editing, C.P., G.R.-P., B.B., L.S. and F.F.; visualization, C.E.; supervision, F.F., L.S. and B.B.; project administration, F.F.; funding acquisition, C.E., B.B., L.S. and F.F.

**Funding:** The study was supported by the Spanish projects TETIS-MED (CGL2014-58127-C3-3-R) and TETIS-CHANGE (RTI2018-093717-B-100), the European project iAqueduct and the Paraguayan Funding Program BECAL.

**Acknowledgments:** The first author is grateful for the good reception received in his stays at the Helmholtz Centre for Environmental Research—UFZ, Leipzig, Germany and the School of Geographical & Earth Sciences, University of Glasgow, Scotland.

**Conflicts of Interest:** The authors declare no conflict of interest. The funders had no role in the design of the study; in the collection, analyses, or interpretation of data; in the writing of the manuscript, or in the decision to publish the results.

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
