# Peer review of "Assessment of Remotely Sensed Near-Surface Soil Moisture for Distributed Eco-Hydrological Model Implementation"

_water, doi:10.3390/w11122613_

Round 1

Reviewer 1 Report

It is a good research, the conclusions are related to the objective of this research, good written. But there are some minor mistakes need to be revised as bellow:

Line 80: rewrite that (https://www.waikatoregion.govt.nz/community/about-the-waikato-region/our-climate/) to avoid plagiarism.

Line 198: It should be 2.5

Line 328: It should be 3.4

There is nothing to be shown in Figure 3, and Figure 4, Figure 9, Figure 14

Reference 14: the question mark should be a dot?

Reference 17: Please remove the parenthesis (brackets) before Stisen, S

Line 527; Reference 28: Please remove the parenthesis before Ahmad.

Line 567: Reference 44: Should show where host that AGU Fall meeting? Check the attachment file

Author Response

Dear Reviewer,

Reviewer 2 Report

The paper is clearly written and suggest real conclusions about soil-moisture evaluation using satellite data.

Few minor modification can be made to improve the work

row 68 where authors explain that 73% of the drainage basin occupied by karst landscapes...here is important to make a reference

also to the soil types row 71-74.

row 137 This sub-model [27,49] has 11 parameters for each type of vegetation cover. Could you to describe all the parameters?

The figures 3 and 4? I can not see the figures no 3 and 4. Also the figure 9 and 14.

The conclusions part can be  synthesized into several phrases without objectives written again.

Author Response

Dear Reviewer, 
